# Effect of Legume Green Manure on Yield Increases of Three Major Crops in China: A Meta-Analysis

**Kailin Liang, Xueqi Wang, Yuntian Du, Guomin Li, Yiqian Wei, Yizhuo Liu, Ziyan Li * and Xiaomin Wei ***

College of Natural Resources and Environment, Northwest A&F University, Xianyang 712100, China; liangkailin@nwafu.edu.cn (K.L.); wangxueqi@nwafu.edu.cn (X.W.); duke.may@foxmail.com (Y.D.); lgmxh@nwafu.edu.cn (G.L.); v1000@nwafu.edu.cn (Y.W.); lyz122324@163.com (Y.L.)
* Correspondence: liziyan0161@126.com (Z.L.); weixiaomin@nwafu.edu.cn (X.W.);
Tel.: +86-151-91858356 (Z.L.); +86-182-92458103 (X.W.)

**Abstract:** The application of legume green manure (LGM) is a traditional and valuable practice for agroecosystem management. In the present study, we conducted a meta-analysis to explore the effect of LGM on the yields of three major grain crops in China under different cropping systems and environmental conditions based on 315 field trial datasets. LGM application increased the yield of the three major grain crops significantly by 12.60% compared to those under no LGM application, with wheat, maize, and rice yields increasing significantly by 9.49%, 16.70%, and 19.22%, respectively. In addition, yield increases were significant under crop rotation with grain crops but not under intercropping. The amount of LGM returned to the field (dry weight) at only 2000–3000 kg/ha and 3000–4000 kg/ha increased yield significantly by 12.32% and 11.94%, respectively. The greatest yield increases were observed when annual precipitation was higher than 600 mm, while annual average temperature was higher than 10 °C, and when soil organic matter content was 0–10 g/kg, with 19.64%, 14.11%, and 32.63% increases, respectively. All regions in China, excluding North China, had significant yield increases, with the largest yield increase, 27.12%, observed in Northeast China. The results of the meta-analysis demonstrated that LGM increases yield of all the three major grain crops in China. Additionally, the benefits were also observed under appropriate planting system, green manure biomass, and environmental factors.

**Keywords:** legume green manure; grain crop; yield; meta-analysis





## 1. Introduction

As the world's population continues to increase, the food production industry is under growing pressure. How to produce more food on our limited arable land is one of the biggest challenges that we will face in the future [1]. Soil is particularly important in the foundation of agricultural production. The intensive misuse of land due to the pressure of increased food demand has often resulted in the degradation of farmland soil. For example, excessive use of chemical fertilizers and pesticides and long-term monocropping have also had a series of negative impacts on soil fertility [2,3]. Therefore, better coordination of food production and ensuring sustainable farming of farmland where the sustainable agricultural management techniques are especially needed have become some of the most serious issues in agriculture today.

In China, the cultivation of green manure (GM) has a long history, and green manure was once an important source of fertilizer to supplement soil nutrients and enhance soil fertility. In 1950, China's GM cultivation area was only 1.7 million hectares. By the 1970s, the area of green manure cultivation was growing rapidly, and the total cultivation area of GM in China reached its peak, with a total area of about 13 million hectares [4]. However, after the 1980s, due to the rapid increase of chemical fertilizer production capacity, people gradually neglected the use of GM in agricultural production [5]. In recent years, the use of GM has been re-evaluated in contemporary agricultural production in response to the rapid

degradation of farmland soils and the pollution of chemical fertilizers, and a great deal of studies on the application of GM have been conducted [6,7]. Legume green manure (LGM) is particularly capable of improving soil fertility due to its well-developed root system and natural nitrogen-fixation capacity [8]. Under the chemical fertilizer reduction policy, legume green manure will be increasingly valued for its excellent soil fertility enhancing ability [9].

The application of LGM, which is an old agricultural management practice, has attracted renewed attention in recent years due to its environmentally friendly properties. LGM could improve soil structure and increase soil organic matter (SOM) contents [10], which are key foundations of soil fertility. LGM of fields enhances soil enzyme activity, increases soil microbial community richness and diversity, accelerates soil nutrient cycling, and suppresses soil-borne diseases [11]. LGM cultivation also has some economic benefits [12].

Crop yield is one of the most critical outcomes in agricultural production, and farmer choices with regard to agricultural management practices are often informed by the effects of such practices on crop yield. Therefore, the effects of LGM on grain crop yield directly influence the probability of farmers to apply LGM in agricultural production. Currently, numerous studies have demonstrated that LGM application significantly increases grain crop yield [12–14]; simultaneously, a few studies have reported that LGM application does not have positive effect on crop yield or even reduces yield sometimes [15–17]. Consequently, in the current context of sustainable agriculture, it is essential to conduct in-depth and extensive studies on the impacts of the application of LGM, which is a traditional agricultural practice to cropland. Currently, most research cases study the effect of LGM on grain crop yield from a single aspect; so, few studies can provide a comprehensive guide for the application of LGM. In view of this, the agricultural industry needs an integrated analysis of the existing independent research data and a comprehensive evaluation of the effects of LGM on crop production in China.

In the present study, field study data from 2000 to 2022 were collected and quantitatively analyzed, with a meta-analysis method, the relationships between crop yield increase and grain crop type, amount of LGM biomass returned to the field (dry weight), planting method, experiment area, and the climate factors. The aim of the present study is to explore the effects of LGM on crop yield under different conditions and to provide a theoretical basis for LGM application in crop agriculture in China.

## 2. Materials and Methods

### 2.1. Data Search and Collection

The keywords "legume green manure", "wheat", "maize", "rice", "yield", and "China" were searched in Chinese and English scientific research databases, such as CNKI, Elsevier, and Science Direct, and field control trial data published from 2000 to February 2022 were collected. The data selected for inclusion in the present study satisfied the following criteria:

(1) The field experiments were conducted in China.
(2) The experimental data were collected from field trials of wheat, maize, and rice cultivation systems.
(3) The field trial designs had a control group with no LGM and a trial group planted with LGM, with other field management practices being consistent.
(4) The data selected had information of trial replicates and mean yield for each trial treatment.
(5) Data from the same experiment appearing in different papers were included only once.

Based on the criteria above, 52 case studies and 315 observations were compiled into the dataset (Figure 1, Table S1). Data displayed in text and tables were extracted directly, while data displayed in graphs were extracted using GetData Graph Digitizer v2.25 (http://www.getdata-graph-digitizer.com, accessed on 1 April 2022). In addition, the number of replicates, standard deviation (SD) of the yield, and other relevant information were extracted (i.e., test sites, planting methods, climatic conditions, etc.).

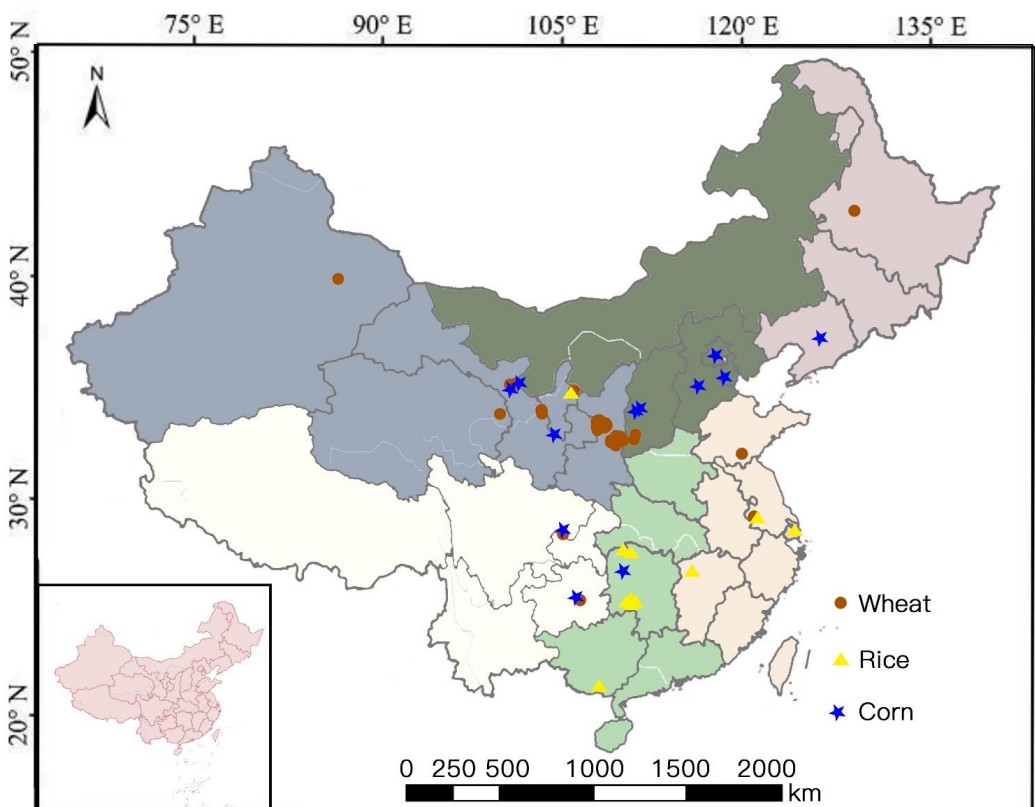

**Figure 1.** Field experiments sites included in this meta-analysis.

The increasing yield effects of LGM were influenced by a series of related factors, and the following influencing factors were collated and grouped according to the relevant trial information extracted from the data: crop type, LGM biomass (dry weight), LGM nitrogen (N) returned to the field, planting method, field experiment location, SOM, soil total nitrogen, soil taxonomy (ST, 1999), annual precipitation, and average annual temperature.

*2.2. Statistical Analysis*

The SD is an essential statistic when performing meta-analyses and is an indicator used to evaluate the weight of each study. When the SD of the yield is provided in the original article, it can be used directly. When the SD is not provided in the original article, but the yield values of multiple replicate tests are listed, the SD is calculated according to the common method. When neither the SD nor the yield values of the test replicates are listed in the original article, but the multi-year test data are included, the multi-year yield data are treated as test replicates and the SD is calculated [18].

The yield means, yield SD, and number of replicates for the control and experimental groups for each study were entered in MetaWin v2.1 (https://metawin.software.informer.com/2.1/, accessed on 15 January 2022), and a random effects model was used to calculate the effect values, ln *R* [19].

$$\ln R = \ln Y_t - \ln Y_c \tag{1}$$

where $Y_t$ and $Y_c$ represent the means of the grain yield of the treatment and CK groups, respectively.

Subsequently, the effect values for each study were evaluated to obtain the overall mean effect value, ln $R_{++}$. The variance $V_i$ and the weight $W_i$ of each independent study need to be determined in the calculation, and the specific formulae are as follows:

$$V_i = \frac{SD_t{}^2}{Y_t N_t} + \frac{SD_c{}^2}{Y_c N_c} \tag{2}$$

$$W_i = \frac{1}{(V_i + \tau^2)} \tag{3}$$

$$\ln R_{++} = \frac{\sum (\ln R_i \times W_i)}{\sum W_i} \tag{4}$$

where $SD_t$ and $SD_c$ were the $SD$s of crop yields in the treatment and CK groups, respectively; $N_t$ and $N_c$ were the number of trial replicates in the treatment and CK groups, respectively; and $\tau^2$ denoted the inter-study variance [20].

To facilitate the interpretation of the effect of LGM application on crop yield, ln $R$ was transformed into yield increase rate, $Z$ [21].

$$Z = (R - 1) \times 100\% \tag{5}$$

If the 95% confidence interval of yield increase, $Z$, is >0, the LGM application had a significant positive effect on crop yield. If it is <0, LGM application had a negative effect on crop yield. If it =0, LGM application had no significant effect on crop yield.

In the present study, a random effects model was used for the data analysis. A Pm < 0.05 indicates that the grouping is significant; conversely, a Pm > 0.05 indicates that the grouping is not significant. Therefore, all subgroups are significant excluding the soil taxonomy (ST, 1999) group (Table 1).

**Table 1.** Results of random effects model inter-group heterogeneity testing.

| Factor | Classification Subgroup | Between Group Heterogeneity | |
|---|---|---|---|
| | | $Q_m$ | $P_m$ |
| Crop type | Wheat<br>Corn<br>Rice | 8.10 | 0.0175 |
| Planting method | Crop rotation<br>Row intercropping | 3.93 | 0.0473 |
| Total biomass<br>(Dry weight, kg/hm$^2$) | <2000<br>2000–3000<br>3000–4000<br>>4000 | 17.58 | 0.0005 |
| N accumulation<br>(kg/hm$^2$) | <50<br>50–75<br>75–100<br>>100 | 13.02 | 0.0046 |
| Test area | China<br>Northwest China<br>Northeast China<br>East China<br>Central and South China<br>North China<br>Southwest China | 17.12 | 0.0089 |
| Soil taxonomy<br>(ST, 1999) | Incepitsols<br>Mollisols<br>Alfisols<br>Ultisols | 0.00 | 0.9999 |
| Soil organic matter<br>(g/kg) | 0-10<br>10–20<br>20–30<br>30 | 28.75 | 0.0000 |
| Soil total N<br>(g/kg) | 0–0.9<br>0.9–1.3<br>>1.3 | 25.65 | 0.0000 |
| Annual precipitation<br>(mm) | ≤450<br>450~600<br>≥600 | 86.51 | 0.0000 |
| Annual average temperature<br>(°C) | ≤10<br>≥10 | 6.13 | 0.0133 |

## 3. Results

### 3.1. Effects of Legume Green Manure on Grain Crop Yield

As shown in Figure 2, LGM significantly increased wheat, rice, and maize yields by 9.49%, 19.22%, and 16.70%, respectively (Figure 2). The rotation of LGM with grain crops increased the yield of grain crops significantly by 7.71%, while intercropping did not increase grain crop yield significantly. When the amounts of LGM returned to the field (dry weight) were 2000–3000 kg/ha and 3000–4000 kg/ha, grain crop yields could be increased significantly by 12.32% and 11.94%, respectively. When the amounts of LGM N returned to the field were 50–75 kg/ha and 75–100 kg/ha, grain crop yield could be increased significantly by 7.93% and 14.39%, respectively. Grain crop yield changed in response to LGM returning to the field, and both exhibited an increase followed by a decrease.

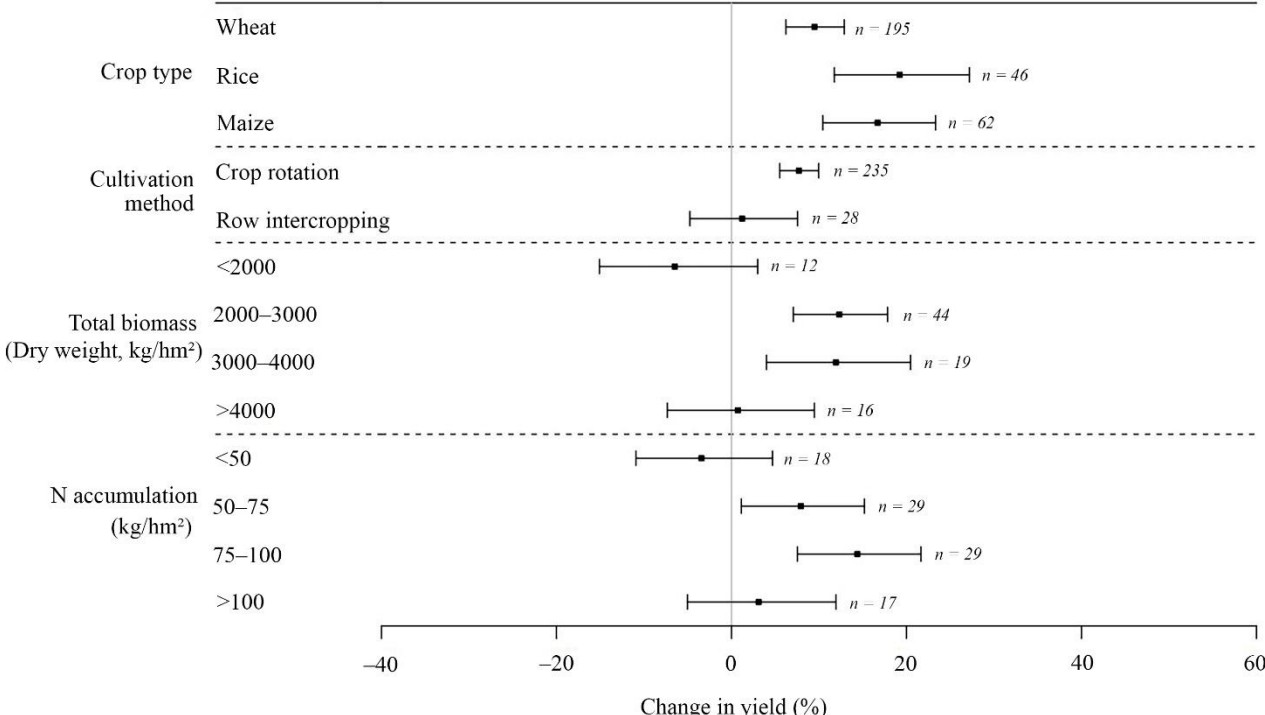

**Figure 2.** Effects of crop type, cropping method, and amount of legume green manure returned to the field on grain crop yield. The n values represent the number of observations. Error bars indicate the effect sizes at $p < 0.05$ level. The effect was statistically significant if the error bar did not bracket the zero graduation. The notes above are also applicable to the rest of the figures.

### 3.2. Effects of Legume Green Manure on Crop Yield in Different Regions and under Different Soil Organic Matter Levels

As shown in Figure 3, within China, LGM showed a significant increase in grain crop yield with an average yield increase of 12.60%. Excluding North China, LGM had significant yield increasing effects in grain crops in all regions of China. The highest yield increase, 27.12%, was observed in Northeast China; the yield increases observed in South Central and Southwest China were not considerably different at 18.59% and 17.73%, respectively. The yield increase in East China was 14.65%, and the lowest yield increase was observed in Northwest China at 11.33%. LGM application had the greatest yield increase effect on grain crops, when the SOM was 0–10 g/kg, with a 32.63% yield increase. Under 10–20 g/kg and 20–30 g/kg SOM, the yield increases were 10.15% and 25.01%, respectively; LGM application no longer increased the yield of grain crops significantly when SOM was >30 g/kg. The maximum crop yield increase of 30.58% was observed at 0.9–1.3 g/kg of total soil N. The crop yield increase was 4.08% and 10.36% at 0–0.9 g/kg and >1.3 g/kg of total soil N, respectively. The overall trend also showed an increase followed by a decrease.

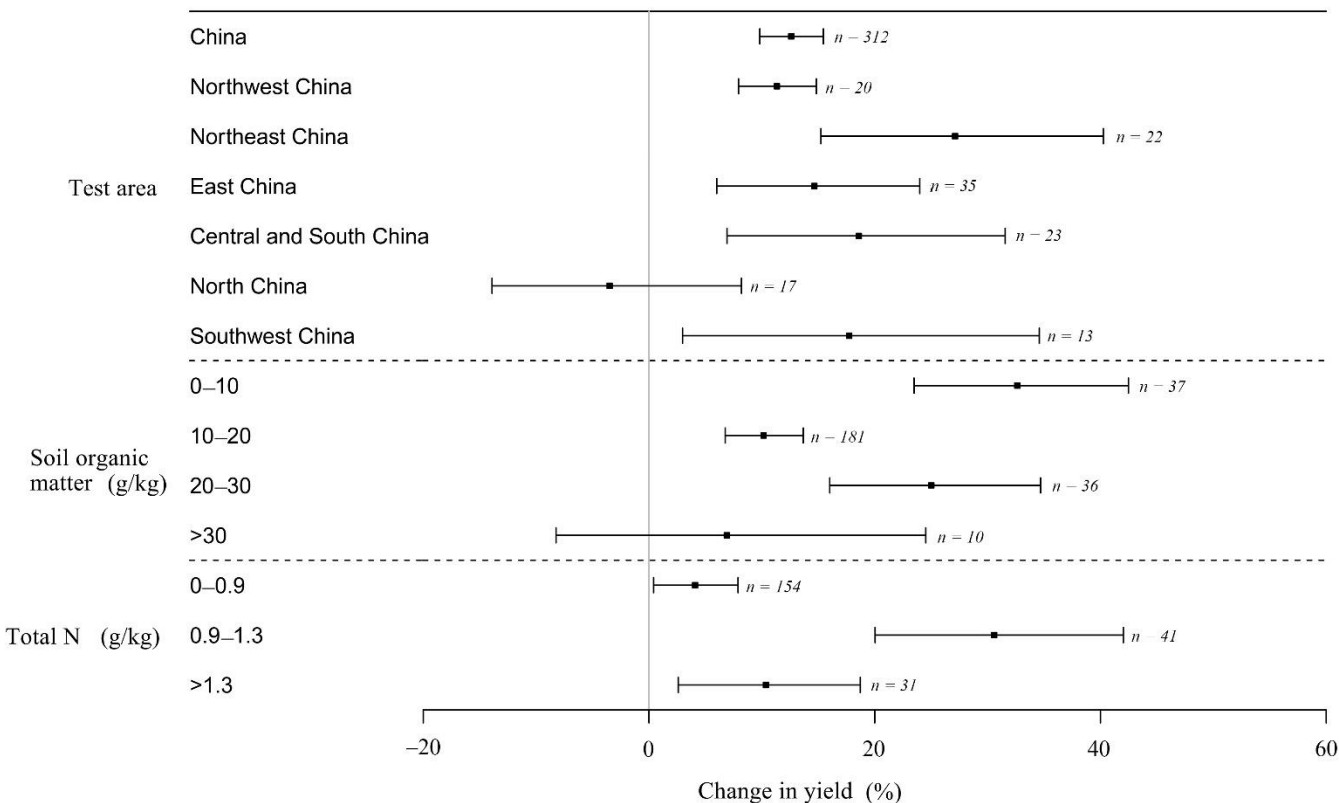

**Figure 3.** Effects of legume green manure on crop yield in different regions and under different soil organic matter levels and soil total N levels.

### 3.3. Effects of Legume Green Manure on Grain Crop Yield under Different Climatic Conditions

Precipitation influenced the yield increasing effects of LGM on grain crops. When annual precipitation was >600 mm, grain crop yield increased significantly by 19.64%. When annual precipitation was 450–600 mm, the effect of LGM on grain crop yield was not significant.

In actual production, wheat cultivation could be divided into spring wheat cultivation (e.g., Wuwei, Gansu) and winter wheat cultivation (e.g., Changwu, Shaanxi). Between them, the spring wheat growing period is concentrated in the rainy season, while the winter wheat growing period is concentrated in the dry season. Therefore, to eliminate the effects of different precipitation levels between the growing periods, the present study also investigated winter wheat in Shaanxi Province, about which there are large amounts of data. As shown in Figure 4, the annual precipitation was divided into four classes: <450 mm, 450–550 mm, 550–650 mm, and >650 mm. Among them, the annual precipitations of 550–650 mm and >650 mm increased the crop yield significantly at 5.82% and 21.2%, respectively. Conversely, under 450–550 mm annual precipitation, crop yield was significantly reduced with a yield reduction rate of 4.83%. Furthermore, the yield increase effect was not significant when annual precipitation was <450 mm, and considering data were only available for nine groups, the results were uncertain. The above results show that the higher the annual precipitation, the higher the rate of yield increase.

As shown in Figure 4, temperature also influenced the yield increasing effect of LGM on grain crops. At an annual average temperature of >10 °C, LGM increased grain crop yield significantly by 14.11%. The rate of increase in yield under an annual average temperature <10 °C was 7.71%, which was also significantly higher but lower than the former.

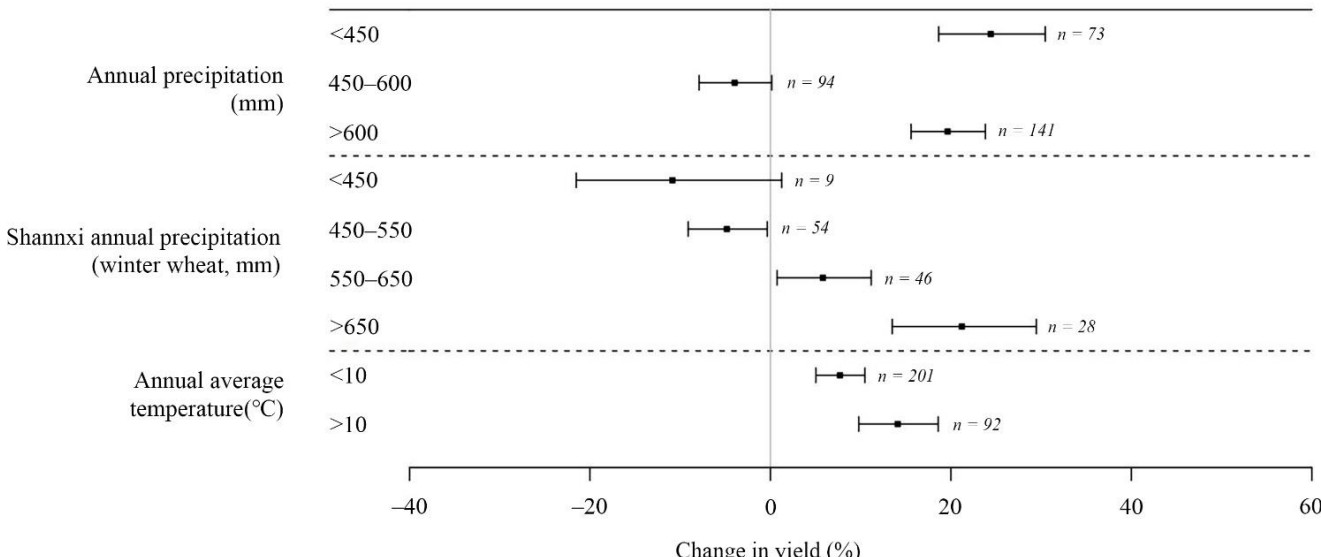

**Figure 4.** Effects of climatic conditions on the yield of grain crops.

## 4. Discussion

### 4.1. Effects of Different Planting Practices and Legume Green Manure Amount Returned to the Field on Grain Crop Yield

LGM application increased the yield of grain crops significantly compared to no LGM application. Among them, the yield increases for maize and rice were similar, which were both higher than the yield increase for wheat, which was only 9.49%. On the one hand, wheat is one of the major crops in northern China, where precipitation is relatively low, and LGM cultivation consumes soil moisture, exacerbating water shortages for wheat growth. On the other hand, northern soils are mostly alkaline, which, combined with low precipitation, can slow down the rates of release of nutrients from returned LGM [22]. In addition, winter wheat growth does not occur in the same periods of rain and drought, which is also one of the factors limiting its yield. In contrast, maize grows in the same periods of rain and drought, and rice growth is not limited by moisture, such that both rice and maize have better growing conditions than wheat.

In the present study, LGM was returned to the field before the sowing of grain crop seeds, regardless of whether the LGM was in rotation or was intercropped with grain crops. In addition, we observed that crop rotation had a greater yield increasing effect on grain crops than intercropping. Therefore, the effect of cropping strategy, in the form of crop rotation or intercropping, on soil moisture may be one of the factors influencing grain crop yield. Under crop rotation, the LGM is returned to the field before the grain crop is sown, and there is no overlap in growth between the two, allowing soil moisture replenishment during the interval. Zhang et al. [23] found that a 13-day interval between green manure tillage and grain crop sowing minimized the negative impacts on soil moisture associated with green manure cultivation. In the case of intercropping, there is a large overlap in growth between the two, and the LGM competes directly with grain crops for soil moisture to the detriment of grain crop growth. However, some studies have shown that intercropping has certain advantages in terms of soil N and phosphorus (P) uptake by grain crops. Soil N uptake by grain crops reduces available N in the rhizosphere of LGM, in turn enhancing N fixation by the legumes to meet their requirements [24,25], which in turn promotes grain crop growth and development. In addition, intercropping LGM with grain crops had been found to improve soil P uptake in several intercropping systems, such as in wheat and white lupin [26] intercropping systems. Competitive and facilitative effects between species always exist simultaneously in intercropping systems [27]. When the magnitude of the competitive effects is lower than that of the facilitative effects, the intercropping has positive impacts. Conversely, when the competitive effects are greater

than the facilitative effects, the intercropping has negative impacts. In addition, the decomposition and nutrient release rates of green manure are relatively slow [16], and crop rotation of LGM with grain crops offers ample time for LGM decomposition, which may also be one of the reasons for the superiority of rotation over intercropping. Furthermore, intercropping can provide organic matter to the soil and improve soil properties; however, such effects may take many years to manifest.

LGM has a high N fixation capacity and can increase soil N contents significantly [8]. In the present study, the LGM return (dry weight) and LGM N return effects on grain crop yield were basically similar, both of which increased followed by decreases. Therefore, the effect of LGM return (dry weight) on grain crop yield may be related to the amount of N returned to the field. The return of appropriate amounts of LGM to the field enhances crop yield. In the present study, the greatest yield increase effects were achieved with 2000–3000 kg/ha LGM return to the field or 75–100 kg/ha N return to the field. When the amount of LGM returned to the field is too large, the excessive N returned to the field would lead to high N metabolism in grain crop stems and leaves, reduced vegetative organ transport, which is not conducive for grain filling, and thus decreased yield [28]. In addition, excessive LGM return to the field would lead to the consumption of large amounts of soil water for growth and increased water competition with grain crops [22], which would in turn adversely affect grain crop yield. In summary, in actual production, the amount of LGM sown and returned to the field should be controlled to minimize the negative effects.

*4.2. Effects of Legume Green Manure on Grain Crop Yield under Different Environments and SOM Levels*

Precipitation has a greater influence on the effects of on grain crop yield than temperature. Planting LGM consumes some soil water, and, when precipitation is low in a year, the soil water deficit caused by planting LGM cannot be offset in time [29]. Therefore, the grain crops grow and develop with scarce water resources, resulting in yield reduction. Conversely, when precipitation is abundant or LGM is planted in paddy fields, the moisture condition is no longer a factor limiting the growth of grain crops, and LGM is able to exert its yield increasing effects on grain crops. In addition, dryland precipitation has a positive effect on LGM decay. Moisture availability can promote the leaching of soluble substances from green manure and LGM decomposition [30]. Furthermore, high soil moisture content can increase microbial populations and enhance microbial activity [31], which in turn promotes decomposition and nutrient release from LGM.

Reducing soil water depletion under dryland green manure cultivation has attracted the attention of researchers in the field of green manure research. Yao et al. [32] and Zhang et al. [33] reported that early harvest of green manure and return as mulch could reduce soil water depletion. In addition, a combination of conservation tillage and green manure cultivation could minimize soil moisture loss due to green manure cultivation [19]. Based on field experiments, Yang et al. [34] and Unger et al. [35] reported that drought years reduced grain crop yield and wet years increased grain crop yields following LGM planting. The trends observed in the present study are consistent with the observations above, where grain crop yield increased with an increase in precipitation after LGM planting.

Temperature is another key climatic factor influencing the effects of LGM on grain crop yield. LGM decomposition requires the participation of a range of microorganisms, and increasing the temperature within certain ranges could increase the soil microbial community richness and enhance enzyme activity [36]. However, Magid et al. [37] observed that low temperature had less effect on the nitrification process and N release following a comparison of N release from green manure at 3 °C, 9 °C, and 25 °C. Therefore, it appeared that the yield increasing effects of LGM caused by increasing temperature are not related to soil N content. LGM cultivation could reduce the ineffective evaporation of large amounts of soil water caused by the absence of vegetative cover and a high temperature environment [38], would increase SOM content, and improve soil physical properties

following return to the field [39–41], enhancing water retention to some degree. Therefore, LGM planting in warmer regions improves water utilization and improves soil properties, which are conducive for increased grain crop production.

In the present study, LGM increased grain crop yields significantly in all regions of China excluding North China. Among them, the highest yield increase was observed in northeastern China, whereas the lowest yield increase was observed in northwestern China. LGM and grain crops in North China were mostly under intercropping and relay strip intercropping, and the numbers of years of planting LGM in the trial were low, resulting in no significant increases in grain crop yields. The temperature in the northeast is lower, and LGM return to the field can enhance soil heat preservation [42]. In addition, the region is rich in agricultural resources [43], and the crop growth conditions are very favorable because of one-year mature and simultaneous rain and heat. LGM application had greater yield increasing effects on grain crops in the south than in the north, excluding the northeast. Precipitation in the south is more likely to be >800 mm [44], whereas that in the north is likely to be <800 mm [45], and sufficient precipitation is favorable for grain crop growth. In addition, the higher temperature in the south is favorable for LGM decomposition. It has been demonstrated that temperature is positively correlated with green manure decay rate, within a certain temperature range [46–48], which may be mainly due to elevated temperatures increasing soil microbial richness and enhancing enzyme activity [36]. In the northwest, light and heat resources are abundant, while precipitation in the region is the lowest compared to precipitation in other regions. Consequently, soil moisture consumption by green manure growth is the main reason for the lowest rate of increase of yield. In conclusion, the regional differences in grain crop yield increase following LGM application are mainly due to soil moisture and temperature factors, which are driven by regional precipitation and temperature.

In addition, in the present study, LGM increased grain crop yield significantly when SOM was <30 g/kg, while the increase was no longer significant at >30 g/kg. The reason may be that, when SOM is low, the appropriate amounts of LGM decomposed after return to the field could increase SOM and make sufficient amounts of nutrients available for the growth of grain crops. Furthermore, when the SOM is high, the organic matter contents could readily meet grain crop growth requirements. Although LGM input to soil would also increase SOM content, the yield increase would not be significant. Similar results have been reported by Zhang et al., following an analysis of green manure cultivation on the Loess Plateau in China [23]. However, their values differed from those in the present study following the adoption of 5 g/kg SOM as the highest limit, which may be attributed to the low SOM contents on the Loess Plateau [49]. The yield increase rate of legume green manure for grain crops also showed a trend of increasing and then decreasing with the increase of soil total nitrogen, which is the same as the effect of soil organic matter on the yield increasing effect of legume green manure. High total soil N may reduce the yield increasing effect of the legume green manure.

## 5. Conclusions

Within China, LGM significantly increased the yield of the three major grain crops by 12.60%, with significant yield increases in all regions excluding North China. The yield increasing effects of LGM under crop rotation were superior to those under intercropping, and the greatest yield increase was achieved in conditions when return amount (dry weight) was 2000–3000 kg/ha. In addition, LGM can lead to significant increases in grain crop yield when SOM is <30 g/kg, whereas the yield increases are no longer significant when SOM is >30 g/kg. The yield increase effect of LGM on grain crops is enhanced with an increase in precipitation along with superior effects during temperatures > 10 °C. In conclusion, LGM can increase the yield of wheat, maize, and rice significantly. Nonetheless, more research is required to determine the yield increasing mechanisms of LGM and provide a rational basis for the reduction of chemical fertilizer application.

**Supplementary Materials:** The following supporting information can be downloaded at: https://www.mdpi.com/article/10.3390/agronomy12081753/s1, Table S1: Information of field experiments in the published references.

**Author Contributions:** K.L. conceived the idea and designed the conceptualization; X.W. (Xueqi Wang), Y.D., G.L., Y.W. and Y.L. performed the data collection and analysis; K.L. and X.W. (Xueqi Wang) wrote the draft of the manuscript guided by X.W. (Xiaomin Wei) and Z.L. contributed to the manuscript enhancement. All authors have read and agreed to the published version of the manuscript.

**Funding:** This work was supported by National Natural Science Foundation of China (42177342), the National Key Research and Development Program (2021YFD9700), Key R & D projects in Shaanxi Province (2022ZDLNY02& 2020zdzx03-02-01), China Agriculture Research System of MOF and MARA (CARS-27), Fundamental Research Funds for the Central Universities (2452021087).

**Institutional Review Board Statement:** Not applicable.

**Informed Consent Statement:** Not applicable.

**Data Availability Statement:** Data are available in a publicly accessible repository. The data presented in this study are available on request from the author.

**Conflicts of Interest:** The authors declare no conflict of interest.

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
