# Peer review of "Effect of Legume Green Manure on Yield Increases of Three Major Crops in China: A Meta-Analysis"

_agronomy, doi:10.3390/agronomy12081753_

Round 1
Author Response
Thank you very much for your valuable suggestions!
The first sentence in the abstract I have replaced with a summary sentence "The application of legume green manure (LGM) is a traditional and valuable practice for agroecosystem management."
To the second question, I would like to answer you here. In my personal opinion, to obtain whether a certain legume green manure is suitable for cultivation under a certain condition requires field trials to get accurate results.
In the course of our study, we found that Chinese milk vetch and ryegrass were often applied to rice cultivation; pea, sweet clover and alfalfa were often applied to maize cultivation; and a variety of legume green manures were applied to wheat. The most suitable legume green manure differs from region to region depending on the conditions available. For example, in Hunan, winter planting of ryegrass in paddy fields is conducive to promoting stability and sustainability of early rice yield, and interplanting of Lathyrus cicera is the best choice for growing legume green manure in the hilly areas of Sichuan. There are many aspects involved in this problem, which is indeed a good one, and I am very willing to attack this problem in my next research.
Thanks again for your suggestions!
Reviewer 2 Report
Dear appreciated Authors,
I have reviewed the article entitled: "Effect of legume green manure on yield increases of three major crops in China: A meta-analysis" and found it very interesting, quality and novel. While the quality of the paper is very good, I suggest minor revision. However, to improve the quality of the manuscript, I propose the following:
Line 19: The greatest yield increases were observed when annual precipitation was higher than 600 mm, while annual average temperature was higher than 10°C and soil organic matter content was 0–10 g/kg, with 19.64%, 14.11%, and 32.63% increases, respectively.
Line 23: The results of the meta-analysis demonstrate that LGM increases yield of all the three major grain crops in China. Additionally, the benefits were also observed under appropriate planting system, green manure biomass, and environmental factors.
Line 36: I would like to suggest just small changes in sentence, as: Therefore, better coordination of food production and ensuring sustainable farming of farmland where the sustainable agricultural management techniques are especially needed has become one of the most serious issues agriculture today.
Line 333: In the northwest, light and heat resources are abundant, while precipitation in the region is the lowest compared to precipitations in other regions.
Line 356: and the greatest yield increase was achieved in conditions when return amount (dry weight) was 2000–3000 kg/ha.
Kind regards,

Author Response
Thank you so much for your corrections to my manuscript!

Reviewer 3 Report
The manuscript is an interesting one and well written. However, according to me there some minor concerns as follows:
1. Abstract should include a line of describing the methodology of the study which is missing here.
2. L39 - 45: Authors need to include some data base about the area under GM crops in globe as well as China.
3. L55-63: Authors must add the research gap here.
4. Objective of your study is not so clear. Need to mention it properly.
5. In results as well as discussion, authors mentioned about the better yield under LGM over no-LGM, which is obvious. However, they did not highlight the scientific background of the increasing the yield using LGM. Authors need to correlate the yield with soil properties.
6. Authors discussed about SOM levels. But, the extent of increase in the SOM using LGM is need to be discussed referring the data of this study. It is lacking here.
7. What is the take-home message from the study?
Author Response
Thank you for your suggestions on my manuscript, I will respond to each of them below.
1.Abstract should include a line of describing the methodology of the study which is missing here.
A: The method used in this paper is meta-analysis, which is mentioned in papers L12-14.
2.L39 - 45: Authors need to include some data base about the area under GM crops in globe as well as China.
A: I have already added " In 1950, China's green manure cultivation area was only 1.7 million hectares. By the 1970s, the area under green manure was growing rapidly, and the total area under green manure in China reached its peak, with a total area of about 13 million hectares. " to papers L41-44, which describe the changes in the area under green manure cultivation of legumes until the 1980s.
3.Authors must add the research gap here.
A: I added the research gap to L68-72 of the paper. The sentence added was " Currently, most research cases study the effect of LGM on grain crop yield from a single aspect, so few studies can provide a comprehensive guide for the application of LGM. In view of this, the agricultural industry needs an integrated analysis of the existing independent research data and a comprehensive evaluation of the effects of LGM on crop production in China."
4.Objective of your study is not so clear. Need to mention it properly.
A: The aim of the present study was to explore the effects of LGM on crop yield under different conditions, and provide a theoretical basis for LGM application in crop agriculture in China. This sentence is in L76-78 of the paper.
5.In results as well as discussion, authors mentioned about the better yield under LGM over no-LGM, which is obvious. However, they did not highlight the scientific background of the increasing the yield using LGM. Authors need to correlate the yield with soil properties.
A: In the results and discussion, I use different temperatures and precipitation as scientific background to explore the increase in yield of legume green manure for grain crops under different conditions. I have also done data previously to classify the data according to the Soil taxonomy (ST, 1999), but the classification was not good (Pm>0.05).
6.Authors discussed about SOM levels. But, the extent of increase in the SOM using LGM is need to be discussed referring the data of this study. It is lacking here.
A: SOM in this study refers to the initial soil organic matter, which is the soil organic matter content before the application of legume green manure.
7.What is the take-home message from the study?
A: In the background of soil environment protection and reduction of chemical fertilizer use, this study explored the factors affecting legume green manure on the yield of three major crops in China, so that farmers can reasonably decide how to apply legume green manure with their own conditions and provide some theoretical basis for the promotion of legume green manure use.
Thank you again for your suggestions!

Reviewer 4 Report
I not see the novelty of this article. Please, note some aspects about this.
Author Response
Currently, most research cases study the effect of LGM on grain crop yield from a single aspect, so few studies can provide a comprehensive guide for the application of LGM. In view of this, the agricultural industry needs an integrated analysis of the existing independent research data and a comprehensive evaluation of the effects of LGM on crop production in China.
In the background of soil environment protection and reduction of chemical fertilizer use, this study explored the factors affecting legume green manure on the yield of three major crops in China, so that farmers can reasonably decide how to apply legume green manure with their own conditions and provide some theoretical basis for the promotion of legume green manure use.